# Gen2Sim: Scaling up Simulation with Generative Models for Robotic Skill Learning

**Abstract:** We propose *Gen*eration to *Sim*ulation (*Gen2Sim*), a method for scaling up robot skill learning in simulation by automatically generating simulation 3D assets, scenes, task definitions, task decompositions and reward functions, capitalizing over large pre-trained generative models of language and images. We propose methods for 3D simulation asset generation from lifting open-world 2D object images using image diffusion models and LLM queries for plausible ranges of physical parameters. We then chain-of-thought prompt LLMs to parse URDF files of generated and human-developed assets to generate task descriptions, task decomposition, and corresponding reward functions, based on the assets and scene affordances. We train reinforcement learning policies in the simulation environments using our generated tasks supervised by the generated reward functions. We demonstrate successful policy learning for a number of long horizon tasks using *Gen2Sim*, without any human involvement. Our work contributes hundreds of simulated assets and tasks for articulated and novel 3D object assets, taking a step towards fully autonomous robotic manipulation skill acquisition in simulation.

**Keywords:** Policy Learning in Simulation, Manipulation, Generative Models

## 1 Introduction

Scaling up training data has been a driving force behind the recent revolutions in language modeling [1], visual understanding [2], speech recognition [3], image generation [4], to name a few. This begs the question: can we scale up robot data to enable a similar revolution in robotic skill learning? One way to scale robot data is in the real world, by having multiple robots self-explore [5] or by collecting kinesthetic demonstrations at scale, with proper instrumentation or crowd-sourcing [6]. This is a promising direction, but safety concerns and wear and tear of the robots might be an obstacle towards autonomous real-world exploration. Another way to scale robot data is in simulation, by scaling up simulated environments and tasks, training robot policies in simulation with reinforcement learning or trajectory optimization, and then transferring them to the real world [7]. Such sim2real paradigm has seen recent successes in robot locomotion [8, 9, 10], in hand manipulation [11, 12], acrobatic flight [13, 14], and deformable object manipulation [15, 16, 17]. However, these examples, though very important and exciting, are still fairly isolated.

A central bottleneck towards scaling up simulation environments and tasks is the laborious manual effort needed for developing the visuals and physics of assets, their arrangement and configurations, the development of task curricula, and reward functions or programmatic demonstrations. In industry, tremendous resources have been invested in developing simulators for autonomous vehicles [18], warehouse robots, articulated objects [19], home environments [20, 21, 22], etc., many of which are proprietary and not open-sourced. Given these considerations, an important question naturally arises: How can we minimize manual effort in simulation development for diverse robotic skill learning?

In this paper, we explore automating the creation of simulation environments, and the development of manipulation tasks and rewards, by exploiting the latest progress in large-scale pre-trained generative

Submitted to TGR Workshop at the 7th Conference on Robot Learning (CoRL 2023). Do not distribute.

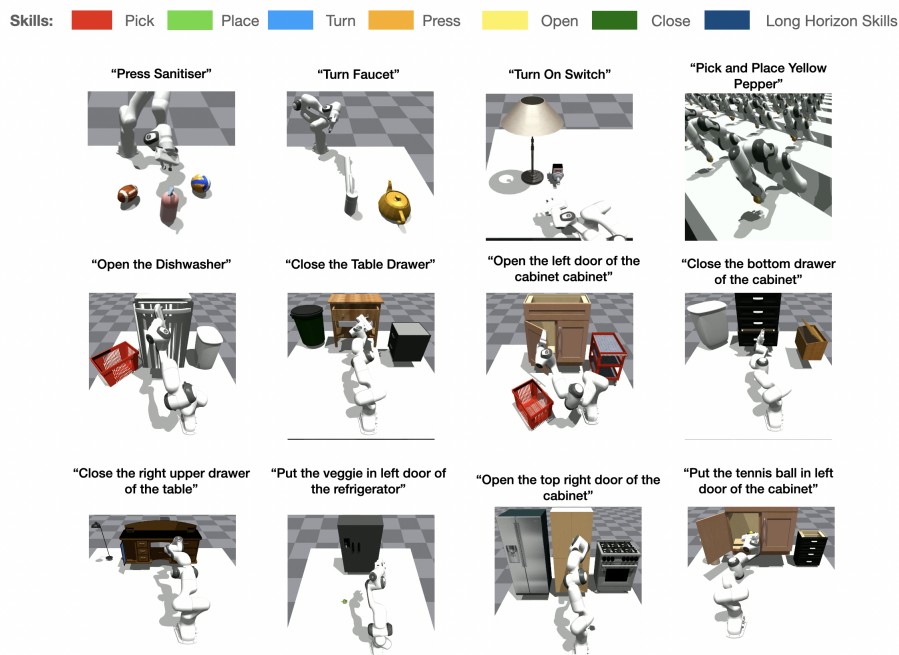

Figure 1: *Gen2Sim* is an automated generative pipeline for assets, textures, physical properties, tasks, task decompositions and corresponding rewards functions, aiming for autonomous robotic skill learning in simulation. Here we show 12 generated tasks, concerning the semantic affordances of the diverse types of assets in the scene.

models, aiming towards automated robotic skill learning. Our system, *Gen2Sim*, strives to automate all stages involved in such development: from low-level 3D assets, textures, and physics properties, to high-level task descriptions and reward functions, leading to automated skill learning in diverse scenarios (See Figure 1). We automate 3D object asset generation by combining image diffusion models for 3D mesh and texture generation, and LLMs for querying physical parameters information. We showcase how LLMs and image generative models can diversify the appearances and behaviors of assets by producing plausible ranges of textures, sizes and physical parameters, achieving "intelligent" domain diversification. We automate task description, task decomposition and reward function generation by few-shot prompting LLMs to predict language descriptions for semantically meaningful tasks, concerning affordances of existing and generated 3D assets, articulated or not, alongside their reward functions. Gen2Sim generates thousands of object assets and task variations without any human involvement beyond several LLM prompt designs. We successfully train RL policies using our auto-generated tasks and reward functions. Last, we demonstrate the usefulness of our simulation-trained policies, by constructing a digital-twin environment of a given real scene, allowing a robot to practice skills in the twin simulator and deploying it back to the real world to execute the task.

In summary, we make the following contributions:

- We show how pre-trained generative models of images and language can help automate 3D asset generation and diversification, task description generation, task decomposition and reward function generation that supports reinforcement learning of long horizon tasks in simulation with minimal human involvement.

- We deploy our method to generate hundreds of assets, and hundreds of manipulation tasks, their decompositions and their reward functions, for both human-developed and automatically generated object assets.

Our code will be made publicly available upon publication. For videos and more qualitative results, see our project site: https://gen2sim.github.io/.

## 2 Related Work

**Large Language Models for task and motion planning in robotics** Large language models (LLMs) map instructions to language subgoals [23, 24, 25, 26] or action programs [27] with appropriate plan-like or program-like prompts. LLMs trained from Internet-scale text have shown impressive zero-shot reasoning capabilities for a variety of downstream language tasks [1] when prompted appropriately, without any weight fine-tuning [28, 29, 30, 31]. LLMs were used to generate task curricula and predict skills to execute in Minecraft worlds [32, 33, 34] Following the seminal work of Code as Policies, many works map language to programs over given skills [35] or hand-designed motion planners [36]. Our work instead maps task descriptions into task decompositions and reward functions, to guide reinforcement learning in simulation, to discover skills that would achieve the generated tasks. Work of [37] also uses language for predicting reward function for robot locomotion, but does not consider task generation and decomposition or interaction with objects. Our work is the first to use LLMs for task decomposition and reward generation, as well as asset generation.

**Automating 3D asset creation with generative models** The traditional process of creating 3D assets typically involves multiple labor-intensive stages, including geometry modeling, shape baking, UV mapping, material creation, texturing and physics parameter estimation, where different software tools and the expertise of skilled artists are often required. It is thus desirable to automate 3D asset generation to automatically generate high-quality assets that support realistic rendering under arbitrary views and have plausible physical behaviours during force application and contacts. The lack of available 3D data and the abundance of 2D image data have stimulated interest in learning 3D models from 2D image generators [38, 39]. The availability of strong 2D image generative models based on diffusion led to high-quality 3D models from text descriptions [40, 41, 42] or single 2D images using the diffusion model as a 2D prior [43, 44, 45]. In this work, instead of a text-conditioned model, we use a view and relative pose conditioned image generative model, which we found to provide better prior for score distillation. Some methods attempt to use videos of assets and differentiable simulations to estimate their physics parameters and/or adapt the simulation environment, in an attempt to close the simulation to reality gap [46, 47, 48]. Our effort is complementary to these works.

**Simulation environments for robotic skill learning** In recent years, improving simulators for robot manipulation has attracted increasingly more attention. Many robotic manipulation environments and benchmarks [49, 50, 19] are built on top of either PyBullet [51] or MuJoCo [52] as their underlying physics engines, which mainly support rigid-body simulation. Recently, environments supporting soft-body manipulation, such as FleX [53], SAPIEN [19], TDW [54], SoftGym [55] and FluidLab [17] provide capabilities for simulating deformable objects and fluids. Our automated asset and task generation are not tied to any specific simulation platforms and can be used with any of them. We unleash the common sense knowledge and reasoning capabilities provided by LLMs and use them to suggest task descriptions, task decompositions, and reward functions. We then use reinforcement learning to discover solution trajectories instead of TAMP-based search.

## 3 Gen2Sim

Gen2Sim generates 3D assets from object images using image diffusion models and predicts physical parameters for them using LLMs (Section 3.1). It then prompts LLMs to generate language task descriptions and corresponding reward functions for each generated or human-developed asset, suitable to their affordances (Section 3.2). Finally, we train RL policies in the generated environments using the generated reward functions, allowing robots to acquire manipulation skills in diverse scenes and tasks. We additionally show the applicability of the simulation trained policy by constructing digital twin environment in simulation, and deploy the trained trajectory in the real world (Section 3.3). See Figure 2 for our method overview.

### 3.1 3D Asset Generation

Gen2Sim automates 3D asset generation by transforming 2D images of objects to textured 3D meshes with plausible physics parameters. The images can be 1) real images taken in the robot's environment,

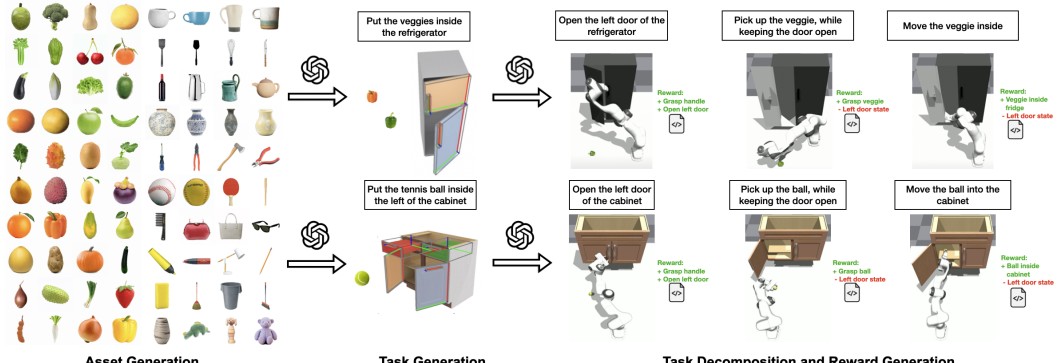

Figure 2: **The Gen2Sim pipeline**: Gen2Sim first generates 3d assets by lifting (generated) 2D images to 3D, and then use both generated assets and assets obtained from other publicly available datasets to populate environments. Afterwards, it queries LLMs to generate meaningful tasks given the scene description, performs task decomposition, generates policy training supervision (reward functions), and yields automated skill learning.

2) real images provided by Google search under relevant category names, e.g., *"avocado"*, or 3) images generated by pre-trained text-conditioned diffusion models, such as stable diffusion [56], prompted appropriately to generate uncluttered images of the relevant objects, e.g., *"an image of an individual avocado"*. We query GPT-4 [57] for a list of object categories relevant for manipulation tasks to search online for or to generate, instead of manually designing it. (Check out our project site for a detailed list of the objects we generated.) Given a real or generated 2D image of an object, we lift it to a 3D model using Score Distillation Sampling [40, 41], initially developed in [40, 58] for text-to-3D lifting. We provide background on image diffusion models below, before we describe our 3D model fitting approach.

### 3.1.1 Image diffusion models

A diffusion model learns to model a probability distribution $p(x)$ by inverting a process that gradually adds noise to the image $x$. The diffusion process is associated with a variance schedule $\{\beta_t \in (0, 1)\}_{t=1}^T$, which defines how much noise is added at each time step. The noisy version of sample $x$ at time $t$ can then be written $x_t = \sqrt{\bar{\alpha}_t}x + \sqrt{1 - \bar{\alpha}_t}\epsilon$ where $\epsilon \sim \mathcal{N}(\mathbf{0}, \mathbf{1})$, is a sample from a Gaussian distribution (with the same dimensionality as $x$), $\alpha_t = 1 - \beta_t$, and $\bar{\alpha}_t = \prod_{i=1}^t \alpha_i$. One then learns a denoising neural network $\hat{\epsilon} = \epsilon_\phi(x_t; t)$ that takes as input the noisy image $x_t$ and the noise level $t$ and tries to predict the noise component $\epsilon$. Diffusion models can be easily extended to draw samples from a distribution $p(x|\mathbf{c})$ conditioned on a prompt $\mathbf{c}$, where $\mathbf{c}$ can be a text description, a camera pose, and image semantic map, *etc* [4, 59, 60]. Conditioning on the prompt can be done by adding $\mathbf{c}$ as an additional input of the network $\epsilon_\phi$. For 3D lifting, we build on Zero-1-to-3 [61], a diffusion model for novel object view synthesis that conditions on an image view of an object and a relative camera rotation around the object to generate plausible images for the target object viewpoint, $\mathbf{c} = [I_1, \pi]$. It is trained on a large collection $\mathcal{D}' = \{(x^i, \mathbf{c}^i)\}_{i=1}^N$ of images paired with views and relative camera orientations as conditioning prompt by minimizing the loss:

$$\mathcal{L}_{\text{diff}}(\phi; \mathcal{D}') = \frac{1}{|\mathcal{D}'|} \sum_{x^i, \mathbf{c}^i \in \mathcal{D}'} ||\epsilon_\phi(\sqrt{\bar{\alpha}_t}x^i + \sqrt{1 - \bar{\alpha}_t}\epsilon, \mathbf{c}^i, t) - \epsilon||^2.$$

### 3.1.2 Image-to-3D Mesh using Score Distillation Sampling

Given an image and relative camera pose 2D diffusion model $p(I|[I_0, \pi])$, we extract from it a 3D rendition of the input image $I_0$, represented by a differential 3D representation using Score Distillation Sampling (SDS). We do so by randomly sampling a camera pose $\pi$, rendering a corresponding view $I_\pi$, assessing the likelihood of the view based on the model $p(I_\pi|[I_0, \pi])$, and updating the differentiable 3D representation to increase the likelihood of the generated view based on the model. Specifically,

the denoiser network is frozen and the 3D model is updated as:

$$\nabla(\theta)\mathcal{L}_{SDS}(\theta; \pi, \mathbf{c}, t) =$$
$$\mathbb{E}_{t,\epsilon}[w(t)(\epsilon_\phi(a_t I + \sigma_t \epsilon; t, \mathbf{c}) - \epsilon) \cdot \nabla_\theta I],$$

where $I = R(\theta, \pi)$ is the image rendered from a given viewpoint $\pi$. We use a two-stage fitting, wherein the first stage an instantNGP NeRF representation [62] is used, similar to RealFusion [43], and in the second stage a mesh-based representation is initialized from the NeRF and finetuned differentiably, similar to Fantasia3D [41]. More information of our score distillation sampling can be found in our website.

SDS was initially developed in [40, 58] for text-to-3D lifting. Gen2Sim considers an image as input for 3D lifting instead. In Section 4, we compare against RealFusion [43] and Fantasia3D [41] that also consider image-to-3D lifting by textual inversion for diffusion adaptation and by an image re-projection loss, respectively. We show our proposed pipeline generates more faithful 3D models from images because the image likelihood provided by the view and pose conditioned image generative model [44] is more informative than a generic or personalized text-conditioned one.

### 3.1.3 Generating plausible physical properties

The visual and collision parameters of an asset are generated from the Image-to-Mesh pipeline discussed above. To define 3D sizes and physics parameters for the generated 3D meshes, we query GPT-4 regarding the range of plausible width, height, and length for each object, and the range of mass given the object category. We then scale the generated 3D mesh based on the produced size range. We feed the mass and 3D mesh information to MeshLab [63] to get the inertia matrix for the asset. Our prompts for querying GPT for mass and 3D object size can be found on our website. We wrap the generated mesh information, its semantic name, as well as the physical parameters into URDF files to be loaded into our simulator.

### 3.2 Task Generation, Decomposition and Reward Function

Given either generated assets or assets obtained from publically available datasets, we prompt LLMs to suggest meaningful manipulation tasks considering their affordances, to decompose these tasks into subtasks when possible, and to generate reward functions for each subtask. We train reinforcement learning policies for each (sub)task using the generated reward functions, and then chain them together to solve long horizon tasks, which would have been impossible without LLMs' decomposition.

Prompts to generate task descriptions, task decompositions and rewards functions contain three elements:

**1. Asset descriptions** We use combinations of assets we generate using the method of Section 3.1, as well as articulated assets from PartNet Mobility [19] and GAPartNet dataset [64]. We populate our simulation environment with randomly sampled assets. Then, we extract information from the URDF files including link names, joints with their types, and limits, using automated scripts. For example, an asset `microwave` has parts [`door`, `handle`, and `body`], and joint [`door-joint`] of type `revolute` with a joint position range $[0, 1]$. We then feed the extracted configurations of the assets to the LLM, with one example shown below:

```
The environment contains the following assets:
1.   asset_name: "microwave"
     part_cofiguration:
         Part 1: "body"
         Part 2: "door"
             - link_name: "link_0"
             - joint_name: "joint_0"
             - joint_type: "revolute"
             - limit: [0, 1]
         Part 3: "handle"
             - link_name: "handle_0"
             - joint_name: "handlejoint_0"
             - joint_type: "fixed"
2.   asset_name: "cup"
     part_cofiguration:
```

```
195        Part 1: cup
196           - link_name: "base"
197           - joint_name: "base_joint"
198           - joint_type: "fixed"
```

**2. Instructions** These are instructions that regulate the response from the LLM. It includes function APIs that can be used by the LLM to query the pose of the robot end-effector, as well as different assets in the given environment:

```
List meaningful manipulation tasks that can be performed
in this environment. Give subtask decomposition and the
order of execution to solve the task. Also, provide the
reward function for each subtask.

The following tasks can be performed in this environment:
1. Open the Microwave Door
2. Close the Microwave Door
3. Pick Cup
4. Place Cup
5. Put the Cup in the Microwave
   This task needs to be decomposed into sub-tasks:
     - Open the Microwave
     - Pick Cup
     - Place the Cup in the Microwave
```

**3. Task and Decomposition Examples** are question-to-language pairs that present few-shot in-context demonstrations of how tasks can be decomposed into subtasks.

```
List meaningful manipulation tasks that can be performed
in this environment. Give subtask decomposition and the
order of execution to solve the task. Also, provide the
reward function for each subtask.

The following tasks can be performed in this environment:
1. Open the Microwave Door
2. Close the Microwave Door
3. Pick Cup
4. Place Cup
5. Put the Cup in the Microwave
   This task needs to be decomposed into sub-tasks:
     - Open the Microwave
     - Pick Cup
     - Place the Cup in the Microwave
```

**4. Examples of reward functions** are task to reward function pairs that present few-shot demonstrations of how tasks can be translated to reward functions. For the following example task, we provide example reward functions composed of 1) distance reward: distance between the end-effector and the target part, and 2) state reward: distance between the current and the target pose of an articulated asset, link, or joint. Note that the following is just an example for the LLM to use as a reference.

```
Task: OpenMicrowaveDoor
Task Description: open the door of the microwave
```
def compute_reward(env):
    # reward function
    door_handle_pose = env.get_pose_by_link_name("microwave", "handle_0")
    gripper_pose = env.get_robot_gripper_pose()
    distance_gripper_to_handle = torch.norm(door_handle_pose - gripper_pose, dim=-1)
    door_state = env.get_state_by_joint_name("microwave", "joint_0")
    cost = distance_gripper_to_handle - door_state
    reward = - cost

    # success condition
    target_door_state = env.get_limits_by_joint_name("microwave", "joint_0")["upper"]
    success = torch.abs(door_state - target_door_state) < 0.1

    return reward, success
```
```

We provide a comprehensive list of our prompts on our website. We show in Section 4 that our method can generalize across assets, suggest diverse and plausible tasks, and reward functions automatically, without any additional human involvement.

### 3.3 Sequential Reinforcement Learning for Long Horizon Tasks

We train reinforcement learning policies using Proximal Policy Optimization (PPO) [65] maximizing the generated reward functions for each subtask. We train RL for each generated subtask in temporal order. Once training for a subtask converges, we proceed to the next subtask. The initial state of the gripper and the environment are sampled from the resulting states of the previous subtask execution. This ensures policies can be temporally chained upon training. Note that while training till convergence doesn't guarantee successful policy training, since we decompose the high-level task into very fine-grained simplistic subtasks, such a heuristic works practically well in our experiments. Our RL policies are trained per environment using privileged information of the simulation state to facilitate learning. Such learned policies can be used as demonstration data and distilled into vision-language transformer policies ([6, 66, 67]), and we leave this to our future work.

## 4 Experiments

Our experiments aim to answer the following questions:

**1.** Can Gen2Sim generate plausible geometry, appearance, and physics for diverse types of objects and parts, without human expertise and with minimal human involvement?

**2.** Can Gen2Sim generate task language goals and reward functions for novel object categories, novel assets with different part configurations, and a combination of multiple assets in an environment?

**3.** Can the generated environments and reward function lead to successful learning of RL policies?

### 4.1 Asset Generation

We compare our image-to-3D lifting with two baselines:

1. *RealFusion* [43], which uses textual inversion of [68] to learn a text embedding for the depicted object concept in an image, and uses text-conditioned diffusion model that uses this text embedding in the text prompt during score distillation.

2. *Make-It-3D* [44], which uses the same NeRF and textured mesh two-stage fitting with SDS as ours do, but does not use a view and pose conditioned generative model, rather a text-based diffusion model, similar to [40].

We show comparisons on several example objects in Figure 3, with images rendered from 4 different views. Our generates more plausible 3D assets as our diffusion prior comes from an image and pose-conditioned model in comparison to approaches like Fantasia3D or RealFusion which uses text conditioning. For more visualizations and details of our generated assets, with diversified textures and their behaviors under gravity and collisions, please refer to our website.

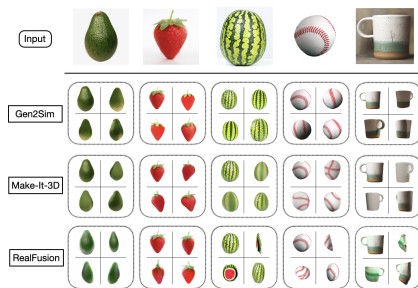

Figure 3: **Left: 3D asset generation** from Gen2Sim, RealFusion [43] and Make-It-3D [44]. Gen2Sim uses a view and camera pose conditioned image generative model during score distillation, which helps generate more accurate 3D geometry in comparison to the baselines.

### 4.2 Automated Skill Learning

We make use of GPU-parallel data sampling in IsaacGym [69] which enables fast and stable convergence of our policies. Our robotic setup uses a Franka Panda arm with a mobile base. It is equipped with either a parallel gripper or a suction cup, depending on the task needs. The suction gripper is only used in pick-and-place tasks where grasping varied geometric objects was intrinsically hard with RL. In all other task categories, we use the parallel jaw gripper. Our state representation for PPO includes the robot's joint position $q \in \mathbb{R}^{11}$, velocity $\dot{q} \in \mathbb{R}^{11}$ (7-DoF arm, $x$ and $y$ for the mobile base and 2 extra DoFs from the gripper jaws), orientation of the gripper $r \in SO(3)$, and poses and

joint configurations of the assets present in the scene. We use position control and at each timestep $t$, our policy produces target gripper pose and configurations, which is then converted to target robot configurations by computing inverse kinematics. A low-level PID torque controller provided by IsaacGym is then used to produce low-level joint torque commands.

Gen2Sim generates diverse tasks, plausible natural language task descriptions, task decompositions and reward functions automatically for hundreds of assets, with different category labels and number of joints, based on the examples provided by the prompt. We show some examples of such generated task descriptions and their corresponding language descriptions in Figure 1 and more on our website. We show example task decompositions in Figure 2. At the time of submission, our pipeline has generated hundreds of tasks, which we will release upon publication. Note that our method can be queried endlessly to generate more tasks and provide task-specific policy demonstrations, which could be used for policy distillation in the future.

We provide all prompts in our website, alongside examples of GPT's responses. Only one example is included in our prompt for task decomposition and reward generation; it concerns the task of "putting a cup in a Microwave". We show then the prediction of GPT4 regarding task and reward function for instances of Door, DishWasher, fruits, veggies and others. Note that the articulation structure structure across all of the assets differ significantly, but our method can effectively generalize. Also, Gen2Sim capitalizes on the common sense knowledge of LLMs regarding object affordances, and thus can produce meaningful ways for interacting with the assets, such as "press the sanitizer" and "turn the faucet", as shown in Figure 1. The rewards generated by GPT can be well optimized with off-the-shelf RL algorithms [65] to learn useful manipulation policies, and the polices are able to solve the tasks upon convergence.

## 5   Limitations

There still remain two limitations that need to be addressed for the proposed system to materialize into a platform for large-scale robot skill learning that are deployable in real-world, as identified below:

**1. Beyond rigid asset generation:** The assets we can currently generate are rigid or mostly rigid objects, which do not deform significantly under external forces. For articulated assets, we are using existing manually designed and labelled datasets ([19, 64]). To generate articulated objects, deformable objects and liquids, accurate fine-grained video perception is required in combination with generative priors to model the temporal dynamics of their geometry and appearance. This is an exciting and challenging direction for future work.

**2. Simulation to reality gap.** Improving fidelity and efficiency of simulators is an active area of research that our method will dramatically benefit from, and we plan to work on. Also, combining explicit physics engines with learnt residual models from simulation and real world alignment [70] for decreasing the simulation to reality gap is an exciting research direction we plan to pursue.

## 6   Conclusion

We have presented Gen2Sim, a pipeline for automating the development of simulation environments, tasks and reward functions with pre-trained generative models of vision and language. We presented methods that create and augment geometry, textures and physics of object assets from single images, parse URDF files of assets, generate task descriptions, decompositions and reward python functions, and train reinforcement learning policies to solve the generated long horizon tasks. Addressing the limitations including generating diverse assets with more complex physical properties, and transfer trained policy to real world using realistic vision input in a closed-loop manner, are direct avenues for our future work. We believe generative models of images and language will play an important role in automating and supersizing robot training data in simulation, and in crossing the sim2real gap, necessary for delivering robot generalists in the real world. Gen2Sim takes one first step in that direction.

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
