# OpenReview forum: "Gen2Sim: Scaling up Simulation with Generative Models for Robotic Skill Learning"
_robot-learning.org/CoRL/2023/Workshop/TGR — CoRL 2023 Workshop TGR Poster_

### Official Review · Reviewer_QYYE · 2023-10-17
**Strong accept**

**Rating:** 9
**Confidence:** 4

**Review:**

Interesting idea to use generative models for scene, task, and reward generation, which further allows RL agent to learn to solve diverse tasks in simulation. Perfect fit to the main topic of this workshop. It would be even better if there are more discussions on how to transfer the learned policy to the real world.

---

### Decision · Program_Chairs · 2023-10-21

Accept (Poster)